# FRIDA, a Framework for Software Design, Applied in the Treatment of Children with Autistic Disorder

Gustavo Eduardo Constain Moreno [1,*], César A. Collazos [2], Susana Bautista [3] and Fernando Moreira [4,5]

1 Escuela de Ciencias Básicas, Tecnología e Ingeniería, Universidad Nacional Abierta y a Distancia, Popayán 190001, Colombia
2 Facultad de Ingeniería Electrónica y Telecomunicaciones, Universidad del Cauca, Popayán 190001, Colombia
3 Escuela Politécnica Superior, Universidad Francisco de Vitoria, 28223 Pozuelo de Alarcón, Spain
4 REMIT, IJP, Universidade Portucalense, 4200-072 Porto, Portugal
5 IEETA, Universidade de Aveiro, Campus Universitário de Santiago, 3810-193 Aveiro, Portugal
* Correspondence: gustavo.constain@unad.edu.co

**Abstract:** The "FRIDA" framework is a guide for the agile development of accessible software for users with Autism Spectrum Disorder (ASD), as a tool for strengthening emotional and social skills in the treatment of autism. It is based on the use of accessible software for the development of emotional and social skills, and designed with a focus on the user with intellectual disabilities. A mixed quasi-experimental study is carried out with three focus groups: children with ASD, expert therapists in ASD treatments and software designers adapting the Design Thinking model for the co-creation of the functional characteristics of the software and its use in therapies. The findings and results show that using FRIDA facilitates the agile design of accessible apps by reducing their development time by 94% and increasing their usability level by more than 90%. This facilitates the treatment of people with ASD, especially in the development of emotional self-recognition skills and social adaptation. The experience applied collaborative design thinking models and agile software design methodologies, articulating knowledge between software developers, therapists, and families of users with ASD. Users were characterized separately, and the functionalities required for the software that would be developed and linked in the treatment of autism were identified.

**Keywords:** autism spectrum disorder; emotional skills; framework; human centered design; recommendations for accessible designs

## 1. Introduction

From the approach of Human-Computer Interaction (HCI) and User Centered Design (UCD), this paper seeks to propose a software framework that contributes to the community of professionals dedicated to the development of computer applications in accessible software design.

There are currently several software programs aimed at users with Autism Spectrum Disorder (which we will name only as ASD) that were evaluated in the project by experts through heuristic evaluation techniques, finding that the vast majority was designed with general accessibility features that are difficult to understand and that the software can't be adapted to the requirements demanded for the therapies of people with diverse characteristics of this disorder.

The contribution of the project is based on the proposal of a collaborative method between therapists and software developers for the identification of the most relevant user requirements for each person with ASD that is characterized, and the agility in the design of the customizable software from the use of a pre-designed and reusable architecture that we call "framework".

The main team of the project was made up of a doctoral student in Electronic Sciences from the University of Cauca in Colombia, who directed the entire project. The consul-



tancies of the project for the technical component were carried out by the Ph.D. student César Collazos from the same university and for Ph.D. student Fernando Moreira from the Universidade Portucalense in Portugal the PhD. Susana Bautista of the Francisco de Vitoria University in Spain. The formation of this team allowed the development of diagnostic practices for the use of the resulting software in clinics and foundations in the cities of Popayán (Colombia), Porto (Portugal) and Madrid (Spain) to validate the impacts achieved.

In the same way, a group of software developers from the University of Cauca and the Siigo company (https://www.siigo.com/ (accessed on 21 July 2021), also from Colombia, were invited for the design tests and the heuristic evaluation of the apps using the built framework.

The clinical team was made up of a group of six therapists from the CENIDI Foundation in Popayán (Colombia) who were directed by Dr. Enrique Valencia, and they were the ones who approved the experimentation activities with children with ASD and validated the impacts on development of emotional skills in children.

With the apps designed from the framework, the aim is to contribute to the development of social skills, typical of emotional intelligence, in children with ASD [1].

In this context, "framework" is understood as a conceptual and computerized support structure for developers, defined with artifacts and specific software modules, which serve as the basis for the organization of user information and the generation of new software programs with accessibility features [2].

ASD is a variation in the neurological conditions of people that manifests itself from their first months of life and intensifies during the first five years [3,4]; we speak of "spectrum" because there are diversity of variations and therefore it is not possible to find two the same cases of suffering from said syndrome [2,5]. The physical origin of this disorder is still under study and therefore therapeutic processes have focused on the development of emotional management skills to improve the quality of life of those who suffer from it [1]. Among the techniques that have most demonstrated their effectiveness are the activities of social behavior management and the emotional self-recognition of the person with ASD carried out using pictograms (graphic representations of objects, people, and daily actions) to improve their communication skills of feelings, emotions and needs [6]. The state of the art carried out shows that, despite the use of manual techniques with pictograms being reliable, they are rarely used through computational tools that motivate the development of skills in a more timely and effective manner [7].

The study that originates this document is based on the identification of an existing need regarding the development of accessible computer applications for cognitive disabilities, especially focused on autistic disorder, where it is important to strengthen the characteristics of social performance of children with ASD: their behavior, interpersonal relationships, emotional communication, and problem solving, as part of their social skills [8]. To do this, an accessible software framework is designed to facilitate collaborative work between therapists and software developers as an important aspect for improving communication and behavior, within social skills in people with autism [1].

The state of the art and the state of the technique carried out in the project and highlighted especially in [1,2,4], demonstrate that the treatment of autism has historically been based on clinical methods for emotional control through medication and behavioral therapies [9]. In recent years, treatment alternatives have been explored for people with ASD from other areas of knowledge [6,8]; for example, from pedagogy through literacy techniques such as GEMMPA [10] that seeks to increase intellectual skills, from psychology with trained pets (dogs or horses) to achieve an emotional leveling of children with intellectual disabilities [11], or from computational sciences with the design of robots [12] or the digitization of cards with pictograms to facilitate communication and emotional management in children with ASD.

The documents studied show that the use of technology adds a positive factor in autism treatment processes by allowing emotional leveling and facilitating learning in children with this disorder [2,7]. However, it is also found that a technological tool that is functional for a person with ASD would not necessarily have the same effect on another

due to the difference in the levels of their suffering. Due to the above, it is important to be able to customize technological solutions according to the cognitive, emotional, and motor characteristics of those who are going to use it.

Initially, the study formulates some recommendations for the design of accessible computer applications to support the treatment of ASD, presented in [1], where the evaluation of the usability factors found in a list of tools is achieved that could be linked to the treatment or formal management of different levels of Autism Spectrum Disorder, the formulation of a user-centered design suggestion base for accessible software focused on said disorder, especially in relation to the design process, the software architecture, and the importance of linking some aspects of gamification in the design of these solutions [13]. Finally, we present the definition of the computational structure of the framework that facilitates the design of software focused on users with this type of cognitive disability.

To give a complete study vision, this paper presents the definition of the problem analyzed, the theoretical findings that support the study, the methodology of the experimental development including the case study for the interaction with ASD children, the collaborative work carried out with therapists and developers of software and the consequent development of the framework that allows the design of accessible software that can be used in therapeutic activities for autism.

## 2. Materials and Methods

This is an interdisciplinary project with a mixed methodological approach that includes a quasi-experimental study through case studies. The activities are organized in three phases: Definition of conditions for the study, Design, and methodological development of the FRIDA framework and finally the Recommendations for the development of accessible software aimed at treating autism (See Figure 1).

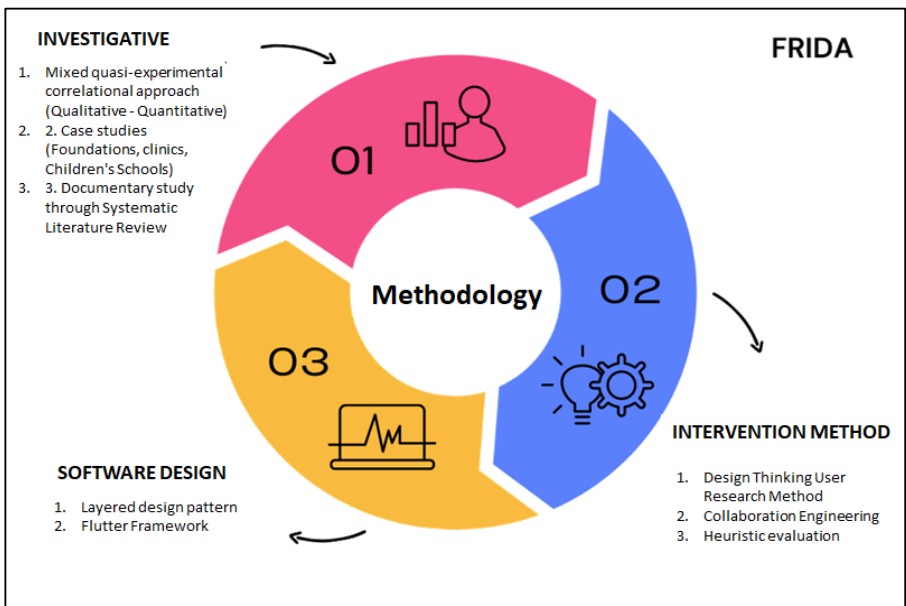

**Figure 1.** Methodological design of the project (Own elaboration).

The activities focused on the design and validation of accessible apps have been directed through Nielsen's usability heuristic evaluation models and instruments [14]. Therefore, all this is validated through metrics and qualitative and quantitative analysis with evaluation indicators of the strengthening of emotional skills in children with ASD and the definition of the characteristics of the software identified with co-design techniques that include the children themselves.

The activities to achieve these purposes are organized in three phases: first, an elaboration of the state of the art (Petersen, 2008), the qualitative analysis of accessible software [15]

for users with ASD and the definition of the software architecture for an accessible software development framework; a second phase of methodological and technological design and development of the FRIDA Framework; and finally a third phase of design of instruments for the validation of the designed framework and verification of the usability of the accessible software aimed at users with autism [1,7].

### 2.1. Phase 1. State of the Art

Within the activities carried out in the first phase, there is the raising of the question of the state of the art, in which the concepts of skills of emotional intelligence have been explored, especially Self-knowledge and Social Skills (Petersen, 2008): the characteristics of the Autism Spectrum Disorder; what treatment and education programs exist and their level of effectiveness [15]; notions of User Centered Design (DCU) and Accessibility [5], including a typology for the existing mobile applications for the treatment of autism; and finally the Metrics and Heuristics that may exist to evaluate the usability of accessible applications [2]. Likewise, an exploration was carried out in the databases of Apple and Google applications for the identification of apps focused on pictographic management for people with ASD and the identification of emotional changes in people through appropriate non-invasive techniques to be applied in children with this condition.

On the other hand, heuristic evaluation metrics were defined to evaluate each of the selected apps in relation to their importance in the treatment of ASD. After a socialization stage about what Autism Spectrum Disorder is, this heuristic evaluation process was applied through evaluation by the following experts for the compiled software:

- An expert in software development;
- An expert in graphic design;
- An expert in Human-Computer Interaction.

Definition of Metrics for the Heuristic Evaluation of Selected Accessible Software

Considering the selection criteria of the mobile applications mentioned in [1], the aim was to evaluate each of the selected apps in relation to their importance in the treatment of ASD. For this reason, the usability criteria were defined that would be applied in the evaluation by experts applied to the compiled software.

At this stage of the project, the usability criteria that inclusive applications should have that are applied within the treatment of autism were defined. These criteria were:

- Ease of Use;
- App Documentation;
- Aesthetic;
- Operability;
- Access to the Tool.

With the criteria already defined, an instrument was built that would allow a heuristic evaluation, based on the concepts of the experts, to be applied to each of the apps selected in the previous activity. This instrument can be detailed in Table 1.

The instrument built for the heuristic evaluation applies a formula to calculate the Usability Percentage (UP) of each of the analyzed applications [7].

$$\text{UP} = \frac{\sum_{i=1}^{i=ncc}(vc * re)}{\sum_{i=1}^{i=ncc}(cc * rc)} * 100$$

where:

*vc*: Value of the usability criterion assigned by the expert;
*re*: Relevance of the usability criterion (automatically calculated by the instrument built for the heuristic evaluation);
*cc*: Number of criteria evaluated;

*rc*: Relevance of the software design criteria with respect to the type of user (person or child with autistic disorder).

**Table 1.** Heuristic evaluation criteria.

| Accessible App Usability Criteria for TEA | |
| --- | --- |
| **Ease of Use** | **Value** |
| 1.1. The representations in the interface are analogous to the real-world aspects. 1.2. The words, phrases, and concepts are familiar and appropriate for the child with ASD. 1.3. The information appears in a logical and natural order. 1.4. The use of images that do not correspond to the real world and that do not contribute to learning (development of emotional and/or social skills) is avoided. 1.5. A consistent and intuitive use is evident in all phases of the application. | |
| **App Documentation** | |
| 2.1. The application presents its own documentation or consultation links aimed at facilitators, therapists, teachers or parents and relatives of children with ASD. 2.2. Contact information for the application developers is presented. | |
| **Aesthetic** | |
| 3.1. The colors of the application have good contrast and are pleasing to the eye of different users. 3.2. The quality of the figures and graphical representations presented are like the counterpart objects in the real world. | |
| **Operability** | |
| 4.1. The application can easily be used by children with ASD according to their motor skills (use of buttons, links, navigation arrows, etc.) | |
| **(Ease) Access to the tool** | |
| 5.1. The software tool is easily accessible through mobile app repositories. 5.2. The application can be downloaded at no cost, at least in its basic version that allows low-cost work in homes and educational institutions. | |

Previously, a desirable value had been established with the experts for the selection of accessible apps, namely, those that obtained an assessment of the relevance of the usability criterion equal to 4 or higher.

This activity determined four mobile applications with the best result of the heuristic evaluation carried out by the experts, whose evaluation can be seen in Figure 2.

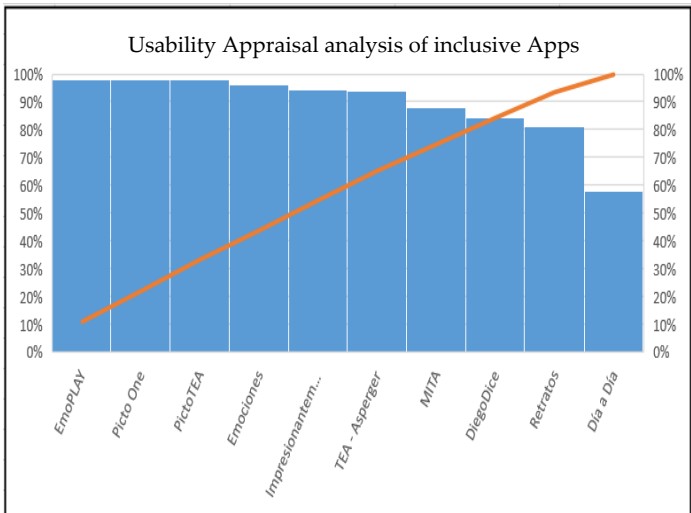

**Figure 2.** Accessible App Usability Analysis.

For the validation of all the findings found above, two case studies are designed in which the user research process is applied from the Design Thinking method proposed by [16] and updated by [17], in which the stages and activities are defined for the characterization of each user who must have accessible software (children with ASD) through the stages of Discovery (usability aspects of the software by ASD users), Interpretation (understanding the information found in the experimental activities with users of software for ASD), Delimitation (of practical ideas for the use of accessible software in treatment processes by expert therapists in the management of ASD) and finally a Validation stage (tqhrough experiences of formal use of the accessible software developments carried out).

The stages of the applied user research process are presented in Figure 3.

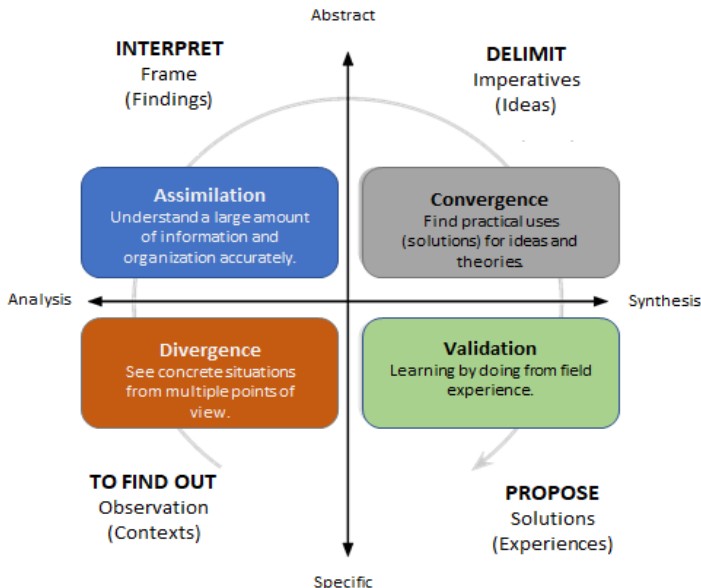

**Figure 3.** Design Thinking User Research Process. Taken from [14].

During the execution of the study, a first case study was designed with two units of analysis (7-year-old girls diagnosed with ASD level 1) so that the effectiveness of the elements analyzed and resulting from the heuristic evaluation of the Apps could be tested . . .
The purpose of the case study was to verify, through human-computer interaction models, whether the use of specialized software for mobile devices allows progress in the results of the treatment of children with ASD by developing some emotional and social skills such as self-recognition and social performance [18].

Due to the types of users contemplated in the study (children with ASD), the application of non-invasive techniques was needed at this stage to assess the effects of linking technology to the therapeutic processes. To do this, the identification of emotions expressed through the face was used through the facial recognition technique allowed with the applications E-motion (developed by the University of North Carolina) and Emotimeter (available in the Google Play Store). In this way, it was possible to recognize the predominant emotions of the software users during their interaction, recognizing for this purpose the movements in the facial muscles.

The use of these tools required the adaptation of the researchers to the therapeutic processes; this was due to the need for acceptance of the challenges posed by children with ASD in the presence of researchers external to the Foundation or Clinic, to achieve our participation in routine activities and facilitate the subsequent use of devices such as smartphones or tablets.

Similarly, the use of the facial emotion recognition technique would have to be verified by the therapists who knew the real emotional expression manifested by each of the children with ASD during each practice. In this sense, the practices were filmed, and photographs were taken that were later analyzed through the Emotimeter or E-motion software to

identify the predominant emotions during each activity carried out with the software, in addition to the identification of the sounds or music that caused greater emotional stability in children. These findings were verified instantly, and also after analysis of the saved images, by the participating clinical team.

One of the emotional assessment results achieved in the first case study carried out can be seen in Figure 4 during one of the interaction sessions with accessible software performed on a girl with ASD.

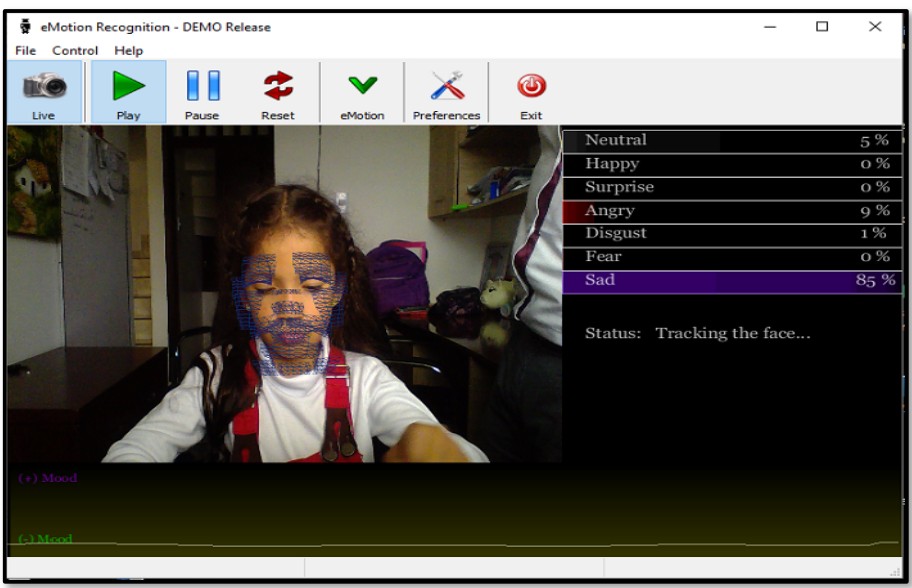

**Figure 4.** Emotions identification through facial recognition [7].

To carry out this activity, a controlled environment was needed, that is, the physical space with all the elements required to document the experiences, such as the provision of a tablet with the pre-installation of the apps best valued by the experts, the possibility of presence of the therapists and a project researcher with a video camera to capture the emotional changes expressed by the child with autistic disorder (accessible software user).

During some sessions of the case study, the emotional reactions that arose during the interaction of the ASD girls with the accessible software were monitored; this allowed the identification of the type of specific activity that caused the changes, in addition to inferring the motor skills that each girl had.

The results of the monitoring of emotional changes using the facial recognition technique (Facial Recognition) showed that working with pictograms and developing activities analogous to the day-to-day activities of the autistic child increases the levels of security, and this is expressed by their facial expressions [18]. In relation to the emotional states during the exercise of using the mobile applications and carrying out the activities proposed in them, each facial reaction was documented through the eMotion application and their mood expressions (Figure 5).

A second subsequent case study (shown in Figure 6), this time with three different units of analysis, sought to revalidate or refute the findings found in the previous study, create a base of previous findings (PRE) for the final validation of the project, and at the same time, include clinical professionals from the Foundation for the Comprehensive Care of People with Intellectual and Cognitive Disabilities CENIDI (https://funcenidi.edu.co (accessed on 17 August 2020) of the City of Popayán (Colombia) in the process of analysis of results of the use of accessible software within the treatment of ASD and at the same time collaboratively design the activities of use of accessible software within traditional therapeutic activities.

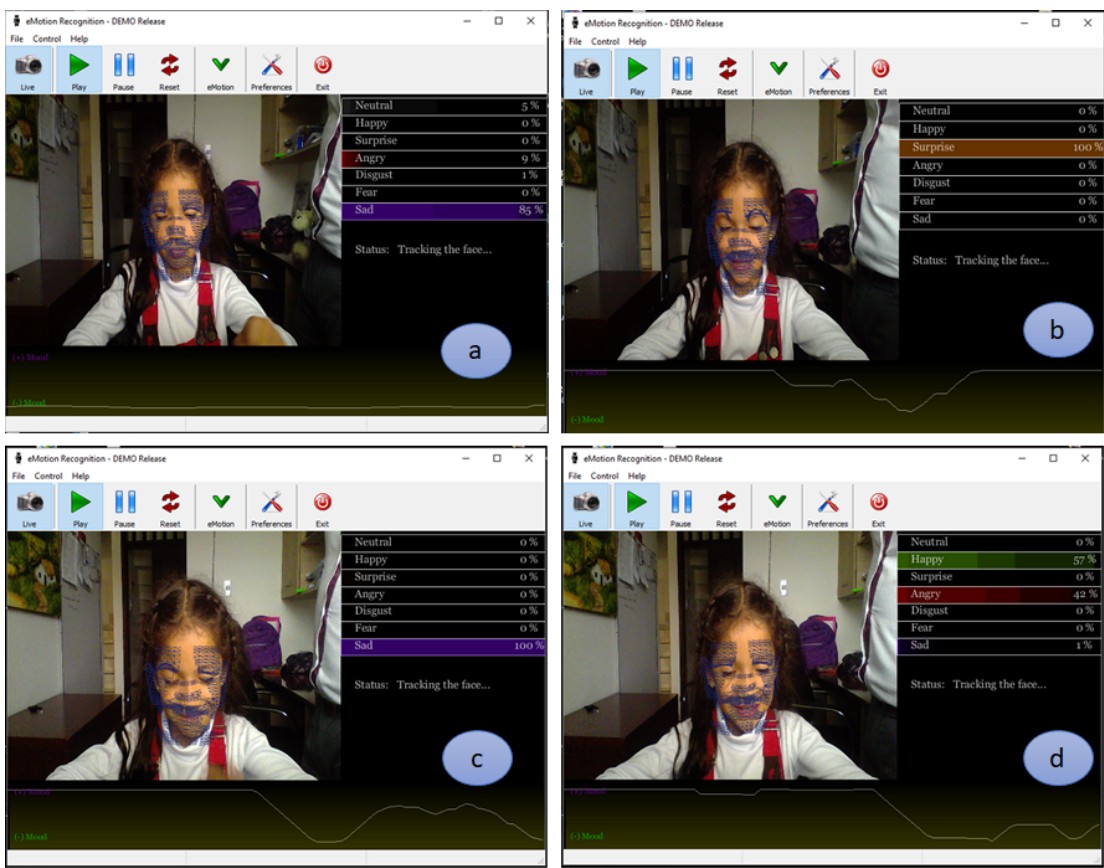

**Figure 5.** Emotion identification using Facial Recognition Case Study 1. Angry (**a**), Surprise (**b**), Sad (**c**), Happy (**d**) (Constain M.G.E. et al., 2020).

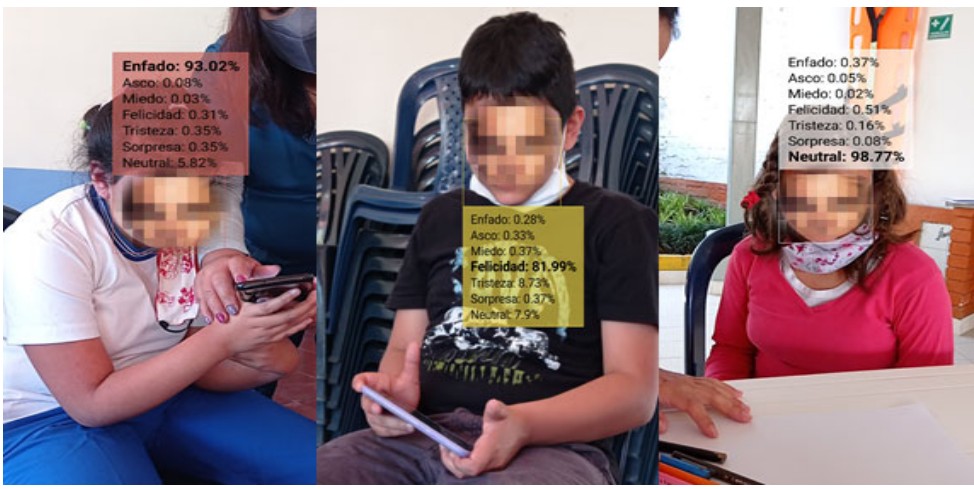

**Figure 6.** Emotion identification using Facial Recognition Case Study 2.

Table 2 correlates the activities carried out with the accessible software used in the experimental practices carried out with ASD children linked to the study and the predominant state of mind in each of them.

**Table 2.** Relationship of predominant emotions with accessible apps use.

| Activity | Predominant Emotion | Evidence |
|---|---|---|
| Visualize pictograms and listen to their pronunciation. | Surprise | Figure 5b |
| Organize sentences based on pictograms. | Sadness | Figure 5a,c |
| Recognize parts of your body expressed graphically. | Surprise | Figure 5b |
| Possibility of expression of feelings, desires, and activities to be carried out. | Happiness | Figure 5b,d |
| Play with avatar and complete tasks | Happiness | Figure 5b,d |

The results of these experiences, which significantly coincide with the results obtained by the experience carried out by [19], allowed us to propose a series of recommendations for the design of accessible software that are valid and apparently effective in the development of emotional and social skills in children diagnosed with different levels of autism spectrum disorder [20]. These recommendations are based on the positive results achieved with the experimental design detailed above and that, according to this, and to the qualitative criteria observed by the children's therapists linked to the two case studies carried out, improve their emotional and social conditions.

Consequently, a framework of recommendations was proposed for the design of accessible software that is valid for use in autistic disorder treatment activities.

### 2.2. Phase 2. FRIDA Framework Development

This stage defines the quality attributes that the accessible software design framework is expected to have, especially in compliance with ISO/IEC 25010 standards [21]. It seeks to obtain characteristics of performance efficiency, compatibility, usability, reliability, security, maintenance, and portability.

As detailed above, the case studies carried out were proposed in two moments: a final moment (POST) where the initial results will be contrasted with those after using the accessible software that is built from the FRIDA Framework; and an initial moment (PRE) for the characterization of existing users with traditional ASD treatment, ensuring the linking of technological devices and the use of accessible software to said processes. This previous activity also included the development of training activities for therapists, linking accessible software to treatment processes and measuring the effects in generating emotional and social changes in ASD children.

Likewise, a methodological adaptation was proposed with respect to the initial case study, this time allowing the articulation of a Human-Centered Design model (methodological adaptation of the classic User-Centered Design (UCD) of the theory of Human-Computer Interaction) with the stages and instruments suggested by the Design Thinking methodology [17] for the co-design of therapeutic activities and the identification of design aspects of accessible software for users with autistic disorder.

In accordance with the purpose of this phase, the aim was to articulate the development of the work, both with ASD children and with therapists from the CENIDI Foundation, also with experts in software development who contribute collaboratively to the design of components and functionality of software applicable to this type of user. The definition of technological aspects, consisting of the design of software modules and the most appropriate architecture for content management in the framework, is carried out through the articulation of Design Thinking stages and an agile software development methodology.

The stage of empathizing, typical of Design Thinking, was carried out in the controlled environment of the CENIDI Foundation and with the permanent accompaniment of the therapists of the ASD children. Various interviews were conducted with therapists and parents to find out the details of the project and its purposes, which were supported by the research ethics code.

The field activities carried out with ASD children (Figure 7), were adapted to the existing therapeutic dynamics, and were carried out in three moments:

(1) Emotional leveling (playful greeting activities and adaptation to the environment and the people present) [22];

(2) Traditional therapy (free drawing and handling of musical instruments) [22];

(3) Technological linkage (access to smart phone devices and tablets with drawing software, management of pictograms for ASD users and monitoring of the most predominant emotional reactions).

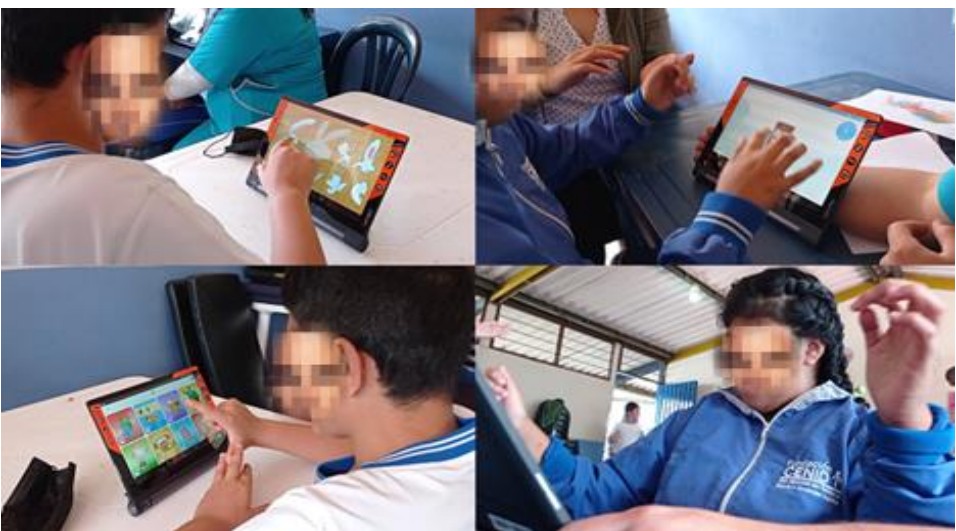

**Figure 7.** Linking accessible software to ASD treatments.

The identification of emotional reactions was achieved with the accompaniment and verification by therapists who have worked for several years with ASD children. Among the most frequent emotions after linking technology to treatment processes are: Neutral State, Joy, and Anger, which have already been identified and presented in Table 2.

This stage allowed the characterization of ASD children (users of accessible software that is built within the project) according to cognitive aspects, emotional and social skills, as well as documented preferences and usability skills in handling accessible software. This is materialized through an empathy map [23], such as the one presented in Figure 8, which was adapted in the study for the characterization of accessible App users (ASD children). With this activity, the data provided by therapists, parents and the children with autistic disorder are collected.

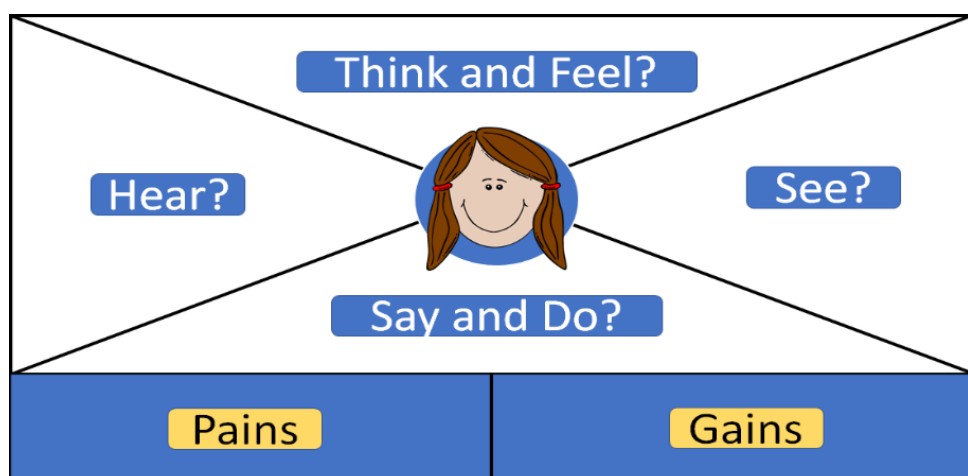

**Figure 8.** Empathy map adaptation for users ASD (Own design).

Likewise, in the Definition and Ideation stage of Design Thinking, it was carried out co-creatively with the therapists, through the techniques of Braindumping (generation of individual ideas) and Brainstorming (generation of group ideas, aloud). The purpose was the definition of functional aspects that accessible software must contain. Part of this activity carried out with the therapists is shown in Figure 9.

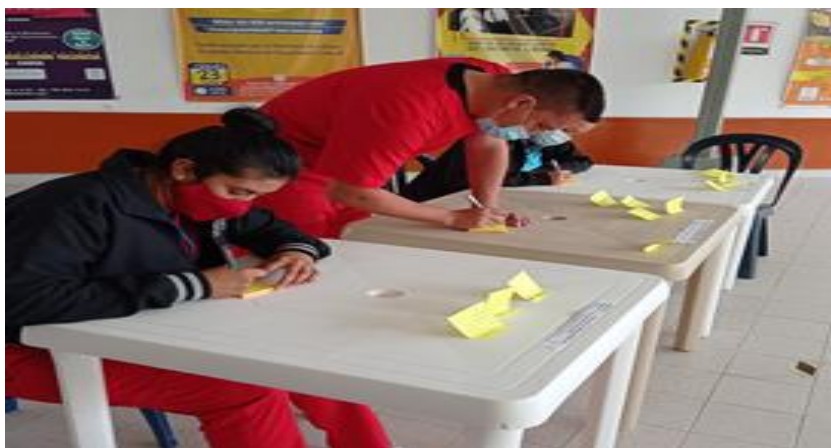

**Figure 9.** Co-creation activities with use of accessible software for users ASD.

With these collected elements, a "translation" was made of the behavioral aspects identified in children with autism into technical requirements that were understandable to software developers. This was done by designing a checklist instrument that presented the functional requirements that an accessible software should contain for each particular user (child with ASD).

Likewise, the next phase of Design Thinking consists in the prototyping of the framework that facilitates the design of accessible software applications that contain the functional characteristics suggested for their usability in therapeutic processes for people with ASD.

The latter was also done co-creatively with software developers from the SIIGO company (https://www.siigo.com/ (accessed on 21 July 2021) in Colombia. The co-design meetings were held on a scheduled basis through Google Meet sessions lasting between 1 and 2 h, in which the design ideas of the components suggested for the FRIDA framework and aspects of software architecture were presented. With the latter, it was sought that the developers make their contributions from their professional experience, giving their opinion on the viability and relevance of the FRIDA Framework within a mass design process of accessible software. This activity has resulted in the microkernel architecture for the FRIDA framework (Figure 10), as well as some plugins that complement the development of each accessible software instance to be built according to the needs or clinical characteristics of the user in which each app comes into focus.

This architecture proposed for the framework contains the modules (plugins) that, according to the previous definitions of the project, can be used to ensure a better use of the accessible software within the treatment of children with ASD. The modules considered for inclusion in the framework are parental control, therapeutic control, level of difficulty of the system, pictogram management, and gamification through rewards. These are in addition to the main modules that allow the expected accessibility features to be inherited, such as: accessibility standards (WCAG, ADA, WAI) [24], subtitles for audio and video, text-to-sound translation, and compatibility according to TEA levels (Figure 11).

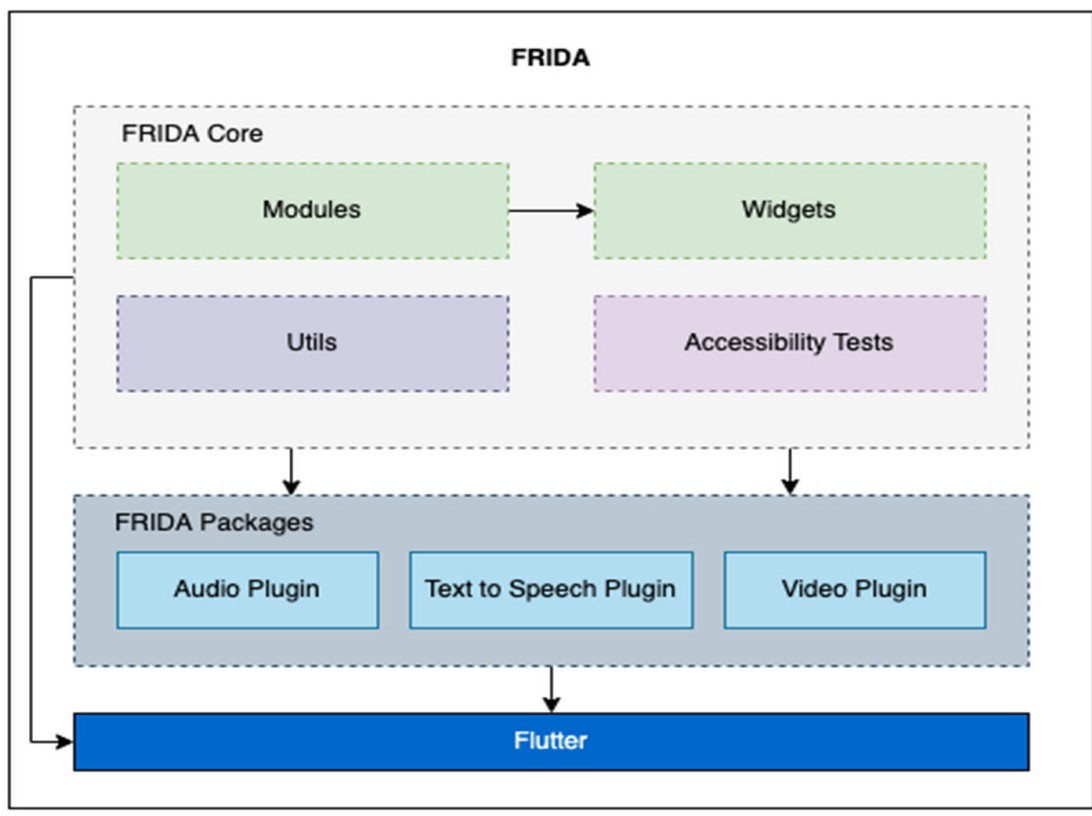

**Figure 10.** FRIDA software architecture (Authors' design).

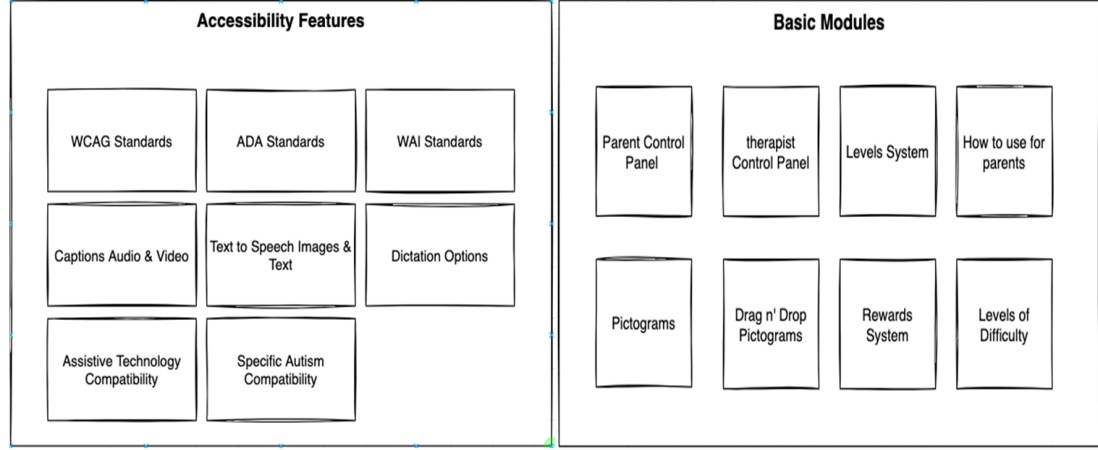

**Figure 11.** FRIDA accessibility characteristics modules (Authors' design).

Finally, during the Prototyping stage, a group of software developers were invited who, using FRIDA, implemented accessible software design instances specially designed for each child with ASD linked to the study. Through online work sessions, the developers learned how to work with FRIDA, received the functional requirements required for each user (child with ASD) and began the implementation of the respective software. Some images of the software applications designed with FRIDA can be seen in Figure 12.

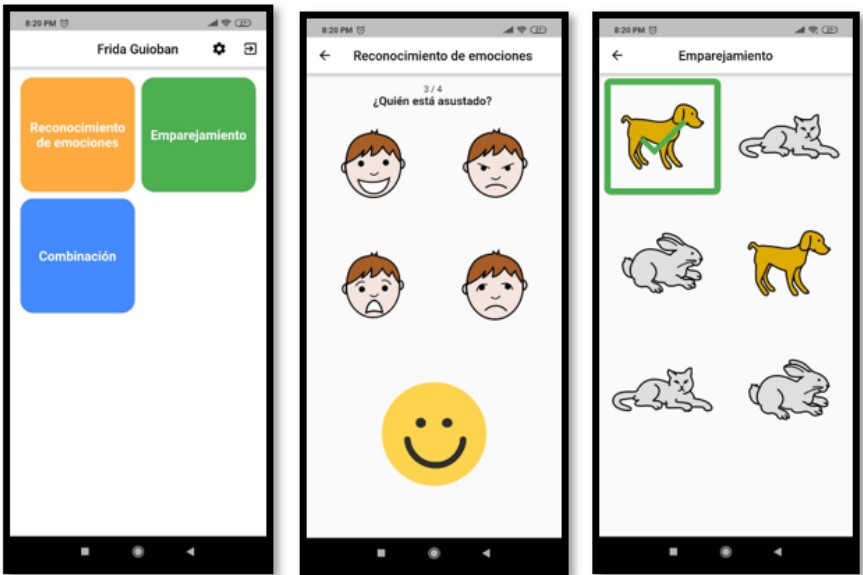

**Figure 12.** App views accessible to users with ASD.

### 2.3. Phase 3. FRIDA's Validation and Accessible Apps

With the FRIDA Framework obtained to meet the need for more computer tools that support the current ASD therapy, several surveys were applied in Spanish-speaking countries (Latin America, Spain and including Portugal). The surveys were aimed at communities of software developers with more than seven years of professional practice and experience in designing software for clinical contexts, to find out their opinion about the importance of linking the development of accessible software for ASD as a line of interest towards which future developments are oriented. Some of the results of this survey can be seen in Figure 13.

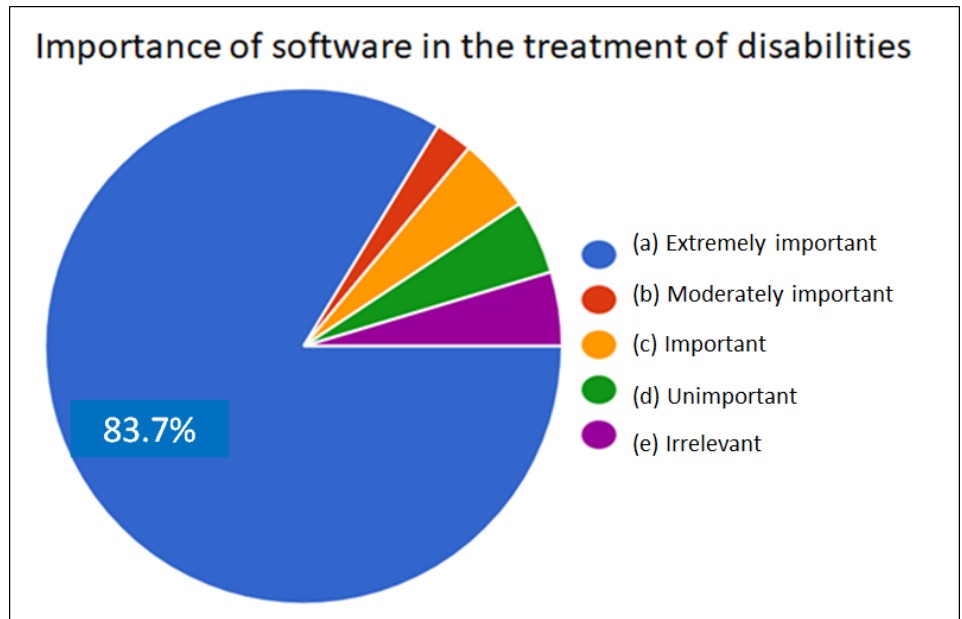

**Figure 13.** Importance of software in the treatment of disabilities (Authors' elaboration).

Knowing the perceived usefulness of the accessible software obtained for the community of software developers, the efficiency and effectiveness of FRIDA and the applications developed with this framework were evaluated. For this, some assessment instruments were designed, both qualitative (surveys aimed at developers and therapists) and quantitative (heuristic evaluation by expert pairs).

The qualitative evaluation carried out by software developers was carried out in two moments:

- Moment 1. A first moment where the characteristics of the Autism Spectrum Disorder and the intellectual characteristics of those who suffer from it are explained; In addition, a document is shared with the functional requirements of a software application for a user with this condition and the time that each developer considers that the design of an appropriate app for this user would take is investigated.
- Moment 2. A video is presented with the characteristics of FRIDA as a tool that speeds up the design of accessible software from the selection of the modules that are necessary for the particular user's requirements and the generation of the app from the framework. After this presentation, the same questions from the previous moment are made so that the developer indicates the time that they consider would be invested in the development of the same application with the use of the already-known FRIDA framework.

Once the survey was applied, the times mentioned by the expert software developers were compared to identify the efficiency of the FRIDA framework. This can be seen in Figure 14.

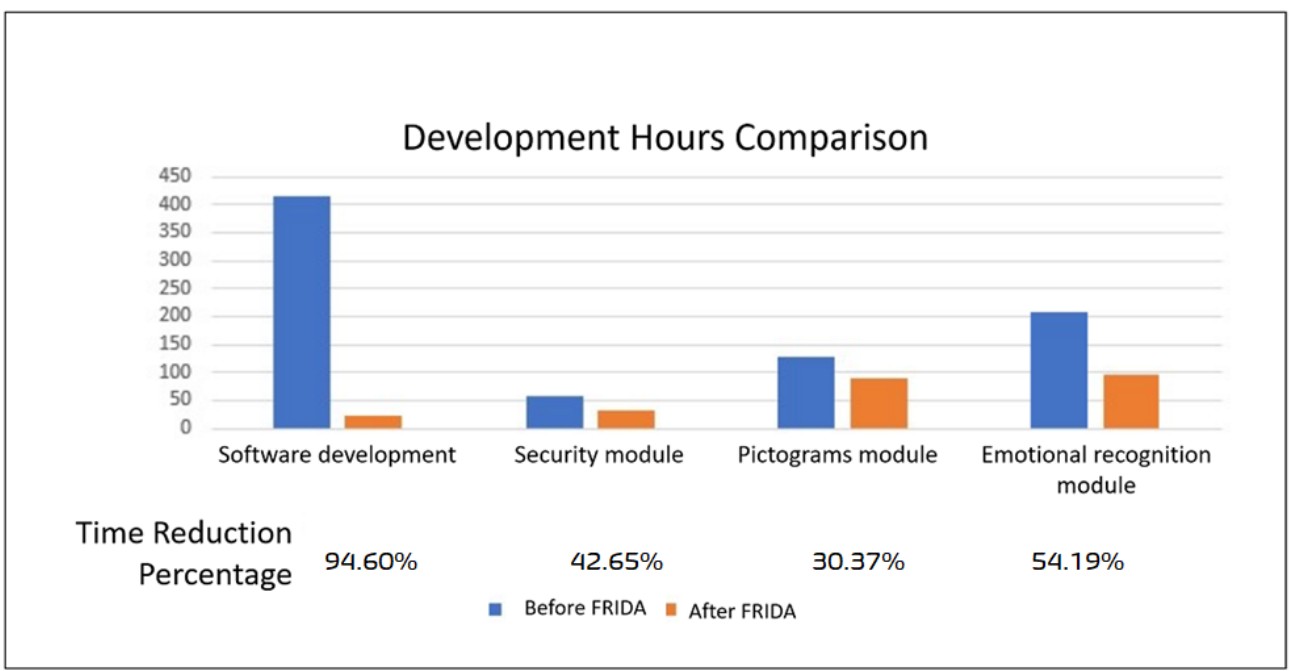

**Figure 14.** Development Hours Comparison (Authors' elaboration).

Similarly, at this stage of the project therapists who are experts in managing ASD are invited. They were then questioned by means of a survey on the positive perception of the use of accessible software in the treatment of autism. Likewise, they were asked about their real intention of linking accessible software in their ASD treatment practices. The results were completely favorable and can be seen in Figure 15.

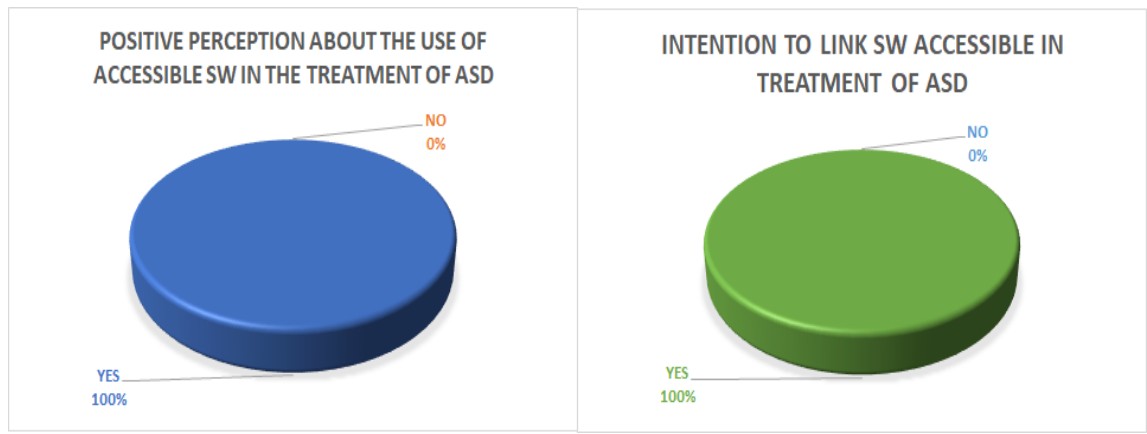

**Figure 15.** Qualitative assessment of expert therapists.

FRIDA's quantitative assessment was carried out with a group of expert developers with more than five years of professional activity. For this, an assessment instrument was designed that was delivered together with the functional requirements of the software for each user linked to the project (children with ASD characterized above). The evaluation criteria (heuristics) defined for this purpose are presented in Table 3.

**Table 3.** Heuristic evaluation criteria by experts.

| Heuristic | Assessment Criteria |
|---|---|
| Heuristic 1: Ease of use of FRIDA | 1. Friendly, familiar, and close base programming language. 2. Base framework documentation accessible in DB or recognized websites. 3. Facility to instantiate accessible apps from the use of FRIDA. |
| Heuristic 2: Accessible App design time | 1. Savings in software design time for reusable modules. |
| Heuristic 3: Access to FRIDA | 1. FRIDA is easily accessible through application repositories for mobile devices. 2. FRIDA is free to download, at least in its basic version, for use by software developers. |

The results of the evaluation of FRIDA carried out by the expert developers can be seen in Figures 16 and 17.

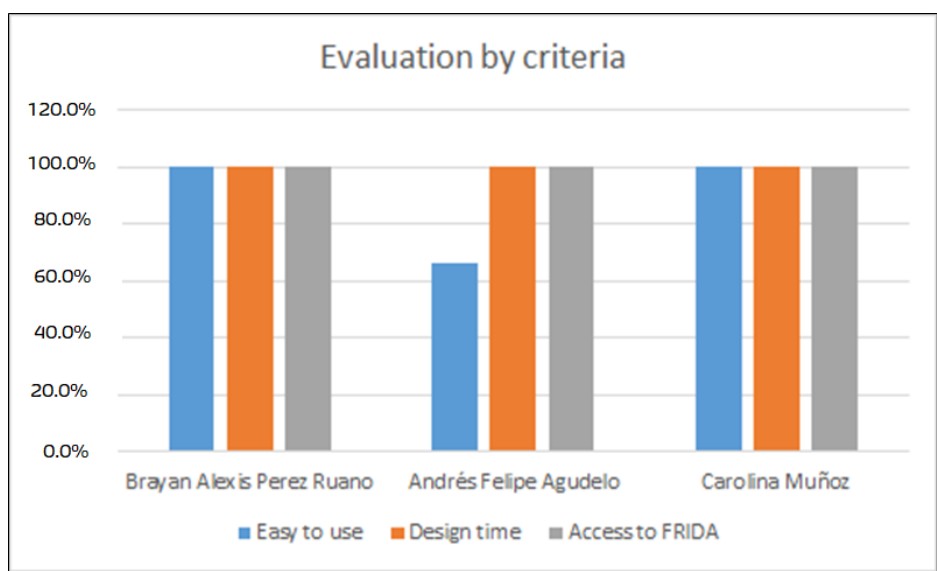

**Figure 16.** Evaluation by criteria.

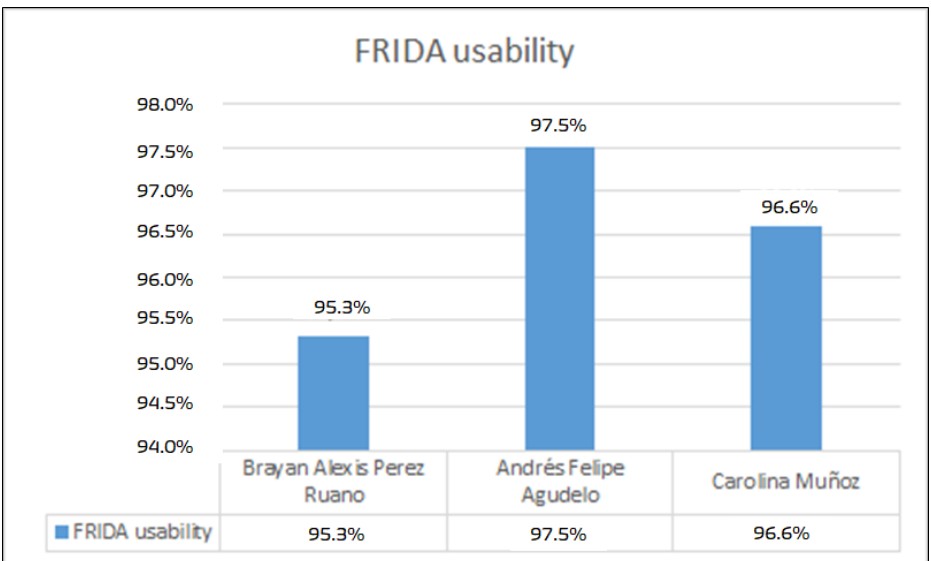

**Figure 17.** FRIDA usability evaluation.

The heuristic evaluation process carried out included the evaluation of the apps designed from the use of FRIDA for three users with ASD. For this, the same instrument built for the heuristic evaluation of the first phase specified in [2] was applied.

This activity obtained the percentage of usability that the software developers assigned to each of the three accessible apps built using FRIDA, this is presented in Figure 18.

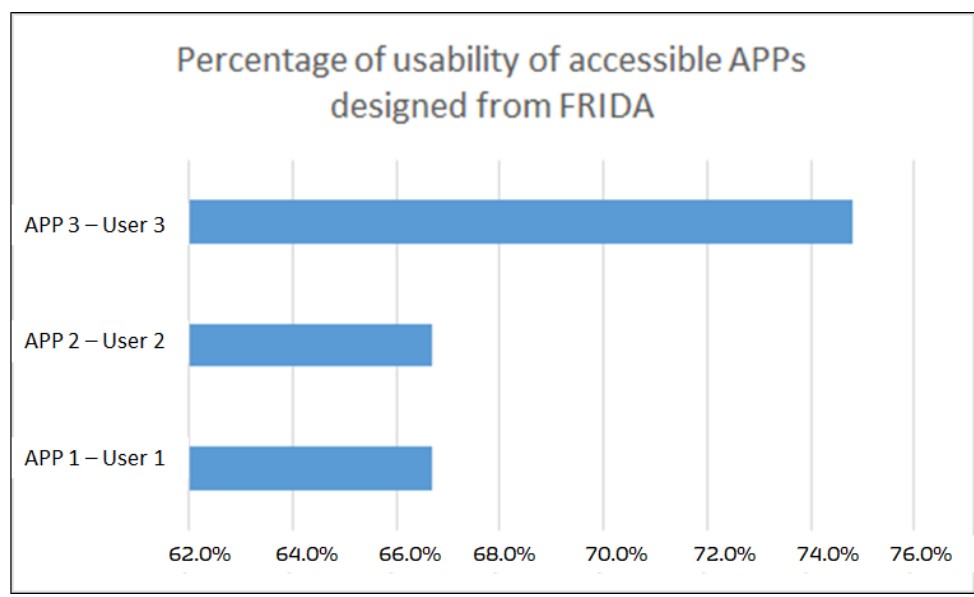

**Figure 18.** Usability percentage of accessible apps designed from FRIDA.

Likewise, the analysis was carried out for each of the heuristics applied in the evaluation related to design aspects of the implemented apps. As shown in Figure 19, the future need to improve the management of colors and contrasts of the built applications is identified, as well as the control of execution errors of some of the included activities.

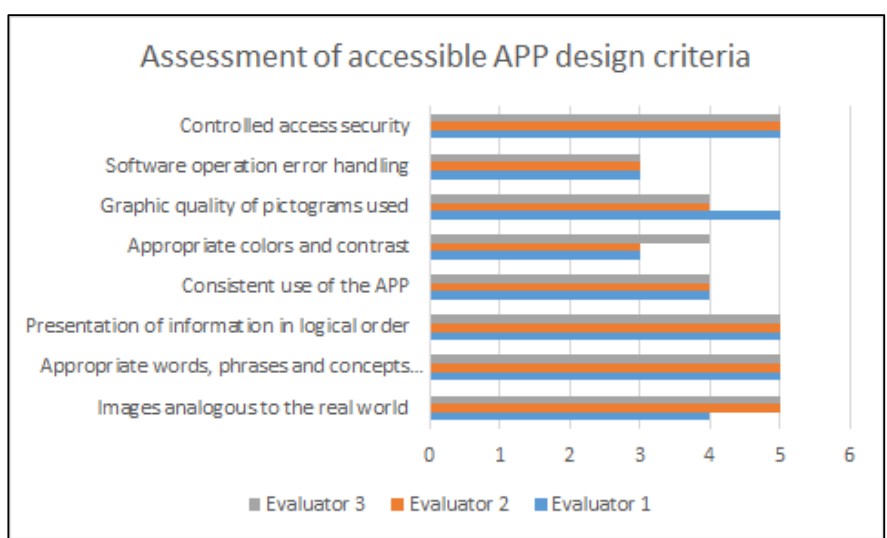

**Figure 19.** Assessment of accessible app design criteria.

## 3. Results

The studies consulted during the systematic literature review stage that are mainly referenced in [1,2] by Constaín et. al., confirm the problem raised at the beginning of this document, specifically, finding that software which has been designed for users with autism is not very useful when its use is extended to people other than the original users. In this sense, the results found in previous experiences could only be compared based on the advantages of using the FRIDA framework with respect to software design processes accessible from traditional development methods. According to what is presented in Figure 14, the use of FRIDA makes the design and development of accessible software aimed at users with ASD more agile, provided that an effective process of identifying user requirements has previously been carried out.

The first phase of the project determines the functional characteristics that accessible software must have to be used in Autism Spectrum Disorder treatments, so that it generates positive results for the development of emotional skills, especially in children within the same age range of the experiment.

However, according to various experiences consulted in the study, the design of the solutions obtained (software in our case) must be tailored to the cognitive and physical conditions of the people who are going to use it (determined by an individual evaluation of each child with ASD) and therefore, the customization of the functions of the software must be possible.

The next phase of the project allowed the adaptation of the Design Thinking model for the collaborative design of the software development framework (working with teams of software developers) and of the therapeutic activities (working with therapists who are experts in treating people with ASD).

Finally, the last part of the project formulates the methods and instruments to evaluate the results of the project from the technological discipline (heuristic evaluation of the framework and the built software) as well as the changes in emotional abilities in children with autism (adaptation of assessment instruments used in conventional therapies performed by therapists).

This research provides a source for the elaboration of new therapeutic instruments, especially accessible software, aimed at communities of software developers for the design of new apps aimed at the personalized improvement of social behavior and emotional management of people with ASD, seeking an improvement in the quality of life, its adaptation and social acceptance.

## 4. Discussion

The linking of technology within the learning processes has proven to be a powerful tool that allows closing the digital divide and accelerating the achievement of learning objectives [3]. In this same sense, the usability of frequently developed software applications that are available in App databases such as those of Google or Apple should be evaluated from their application context and not only from their software architecture. It is for the above reasons that usability should be considered as a software quality attribute that evaluates how easy it is to use the graphical interface of computer programs, especially when its users are people with some type of disability.

Based on the above, the primary notion of usability is that software designed from the psychology and physiology of the user is more efficient to use, since it would take less time to perform a particular task, easier to learn because an operation can be understood by merely looking at the objects present in the graphical interface, and moreover, more satisfying to use [3]. The foregoing becomes more important when working with users suffering from cognitive disabilities that, such as Autism Spectrum Disorder limit not only their cognitive abilities but in many cases are accompanied by physical limitations.

This is where it is evident that the classic models of software design can be used for the development of software aimed at users with disabilities, but where their impact and effectiveness are not sufficient because they ignore important aspects of the characteristics of end users [18]. In principle, technological improvements and, above all, the growing competition between software developers, mean that the needs of users are taken more into account every day from the very moment of the conception of a product [25]; however, the solutions adopted also end up benefiting sectors of the population with disabilities in a generalized way but omitting specific needs.

In the case of people suffering from Autism Spectrum Disorder, Neuropsychological Rehabilitation has been one of the most important instruments in the treatment of alterations of higher cognitive functions (Attention, Memory, Perception, Orientation, Verbal Learning, Calculus) [25]. For these cases, there are simple evaluation and rehabilitation programs for people with cognitive deficits or impairments, through direct user interaction using multimedia systems and touch screens, without the need to use other devices such as keyboards, trackball, or the mouse, which in some cases allow the reversal of cognitive decline and recovery of some higher brain functions [26]. However, in the case of autistic disorder, considerations are greater when designing software due to the elements of the same therapeutic and clinical process that must be considered if a greater and more effective impact on rehabilitation is expected.

In the same sense as mentioned above, the literature review carried out in the study found that a good number of software resources had been developed. Despite this, there is little software that is currently in use, because the foundations that work with people with disabilities are not aware of these initiatives, there are no alternatives that allow them to be socialized and promoted, and to a large extent because these developments are built on general form for a specific disability, or directed towards people in particular conditions that can hardly be cross-applied to others whose condition of cognitive disability may seem similar.

As a turning point in the face of the situation manifested, the FRIDA Framework is proposed as a guideline framework in the design of software with accessibility features that allows software developers to work in an articulated manner with therapists who work with patients diagnosed with Autistic Spectrum Disorder. If it is taken into account that from the case studies carried out in the project, the advantage of using mobile devices is evident, over the desktop PC, for the interaction of ASD users and the computer applications created for the development of skills in people with special needs, a mechanism is required that facilitates the design of software tailored to the specific requirements for each particular case (person with autism in the process of treatment). Therefore, the reference framework provided by the FRIDA Framework works best when collaborative work has previously been carried out between software developers and therapists for the correct characterization of the end-users who will use said applications.

## 5. Conclusions

The design of software with accessibility features aimed at users with ASD is facilitated by the existence of a framework of recommendations for the identification of user requirements (suggested characterization through the Design Thinking user research method), while the use of the FRIDA Framework generates the software architecture for each legacy application and streamlines the configuration of said software according to the details of each particular user. Interdisciplinary interaction between therapists and software developers will therefore always be recommended to achieve better results.

The social awareness of the FRIDA initiative in different Latin American countries, and other countries with Spanish-speaking professionals, demonstrates the interest in its use by therapeutic communities that work with ASD due to the innovation demonstrated in the processes of technology use. In addition, it was possible to generate interest in the achievements regarding the development of therapeutic tools that contribute to the strengthening of emotional and social skills in children with this disorder. Similarly, the software development communities have shown interest in linking accessible software as one of their current lines of work and with the market potential for such products. Even parents of children with ASD have expressed the need to have these developments to improve the quality of life of these people through emotional improvement and levels of social interaction.

The foregoing has been compiled through open surveys shared between communities of expert therapists in the management of ASD, relatives of people with ASD, and software developers, with all of whom the impacts of the use of FRIDA on people with autism continue to be validated.

The development of emotional and social skills in people with ASD is assisted by linking the use of software with accessibility features to their therapeutic process, one which is configured for the cognitive conditions and usability possibilities of said user. The foregoing must be guaranteed through procedures for characterizing each user, so that the cognitive abilities of the person with ASD are identified and the accessibility elements that can best transmit emotional development from the therapeutic planning carried out by clinical experts.

The research was carried out under case study models with different units of analysis (children between 7 and 17 years old at the time of the study) with whom various activities adapted from the Design Thinking model were worked on, the recreational activities of treatment of the ARASAAC project, and co-design processes of the software architecture for the FRIDA Framework that is proposed and tested within the activities described in this document. To obtain a greater complement of the results achieved, it is recommended that the experience be replicated with various cases of children with Autism Disorder following the steps of prior characterization of their conditions of cognitive disability, the usability possibilities of technological devices (Tablet or smartphones) and the accompaniment of experts in the management of ASD who validate the emotional and social changes achieved at the end of each user experience.

The environment of the child (ASD user), the level of knowledge of the parents about the ASD and their commitment to the child, and the ways they use to handle the situation should be evaluated. The foregoing serves to characterize the functions that software that is designed as a therapeutic instrument for each user (child with ASD) should contain. In this sense, it is important to monitor the proposed framework of recommendations, together with the use of the FRIDA framework, to obtain software with accessibility features that achieve better results in existing therapeutic initiatives.

Finally, it is expected to continue validating the FRIDA framework with other people diagnosed with ASD from different countries to increase its possibilities of designing accessible software and its application in treatment processes for this syndrome, as well as continuing to deepen the strategy for the development of emotional or social skills in other types of cognitive disability.

As for future works in this same line of technology and health, it is expected that efforts will continue to delve into the use of technology to identify greater cognitive aspects of people with ASD. Among the aspects of knowledge to deepen is the neuronal functioning of children with ASD when using software and its influence on emotional reactions, in addition to extending the experience presented in this paper to people with other intellectual disabilities such as Asperger's or Attention Deficit Disorder ADD.

**Author Contributions:** Conceptualization, S.B.; Data curation, F.M.; Investigation, G.E.C.M. and C.A.C.; Methodology, G.E.C.M.; Supervision, C.A.C., F.M. and S.B.; Writing—original draft, G.E.C.M.; Writing—review & editing, C.A.C., S.B. and F.M. All authors have read and agreed to the published version of the manuscript.

**Funding:** This work was supported by the FCT—Fundação para a Ciência e a Tecnologia, I.P. [Project UIDB/05105/2020].

**Institutional Review Board Statement:** The study was approved by the Program Committee of the Doctorate in Electronic Sciences of the University of Cauca in Colombia (approval document dated 9 February 2022).

**Informed Consent Statement:** Informed consent was obtained from all subjects involved in the study.

**Data Availability Statement:** The study did not report any data.

**Conflicts of Interest:** The authors declare no conflict of interest.

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
