# Peer review of "FRIDA, a Framework for Software Design, Applied in the Treatment of Children with Autistic Disorder"

_sustainability, doi:10.3390/su142114560_

Round 1

Reviewer 1 Report

Dear Editor, 

I have carefully assessed this manuscript. The topic is of interest in the field of ASD and neurodevelopmental disorders in general. The presentation of the work, despite being extremely original, is appropriate for the type of the study.

My main concern regards the connection between the existing background, the working hypotheses and the aims of the study. Given the marked novelty of the work, a specific section should be provided, to collocate this intervention as a solution for existing gaps in the literature, not just a good idea. 

Author Response

Dear #1 Reviewer

Sustainability Magazine

Reviewer 2 Report

1.      Its existing title's capitalization should be updated to follow the MDPI format.

2.      The authors need to provide all of the emails after affiliation except for corresponding authors based on MDPI format.

3.      The abstract requires the addition of quantitative results.

4.      Please conclude your abstract with a "take-home" message.

5.      Sort the keywords according to alphabetical order.

6.      Make the each of keywords with lowercase font following MDPI format, revise it.

7.      What is the current study's novel? It has been extensively researched in the past. Nothing truly novel in its current state. The absence of anything original makes the current study seem like a replication or a modified study. The introduction section should contain specifics about the writers' uniqueness. It is a significant reason to reject this study.

8.      In order to highlight the gaps in the literature that the most recent research aims to fill, it is crucial to review the benefits, novelty, and limitations of earlier studies in the introduction.

9.      In addition to using applications to treat children with autism spectrum disorders, the use of therapeutic tools has also been used as reported by Afif et al. by using a hug machine with the concept of deep pressure which gives a calming effect. It is a vital topic that authors must provide in the introduction section. Additionally, the MDPI's suggested reverence should be taken to substantiate this explanation as follows: Afif, I. Y.; Manik, A. R.; Munthe, K.; Maula, M. I.; Ammarullah, M. I.; Jamari, J.; Winarni, T. I. Physiological Effect of Deep Pressure in Reducing Anxiety of Children with ASD during Traveling: A Public Transportation Setting. Bioengineering 2022, 9, 157. https://doi.org/10.3390/bioengineering9040157

10.   In order to improve the reader's understanding of the materials and methods section simpler, the authors could provide a figures that clarify the workflow of the current study rather than only the predominant text as it currently appears.

11.   The inaccuracy and tolerance of the equipment used in this inquiry are critical details that must be included in the article.

12.   The error and tolerance of the experimental tools used in this investigation are important aspects that have to be mentioned in the manuscript. It might be valuable for further research by other scholars because of the different results.

13.   Outcomes must be compared to similar past research.

14.   What is the limitation of the present work? Please include it before the conclusion section.

15.   Mention further research in the conclusion section.

16.   The reference should be enriched with literature from the last five years. Literature published by MDPI is strongly recommended.

17.   Because of grammatical faults and linguistic style, the authors must proofread the document. MDPI English editing service would be a solution.

18.   Please ensure that the authors followed the MDPI format correctly; modify the current form and recheck, as well as any other problems that have been highlighted.

Author Response

Dear #2 Reviewer

Sustainability Magazine

Round 2

Reviewer 2 Report

Well work by authors.